# Omega-3 Polyunsaturated Fatty Acids EPA and DHA as an Adjunct to Non-Surgical Treatment of Periodontitis: A Randomized Clinical Trial

**DOI:** 10.3390/nu12092614

**Published:** 2020-08-27

**Authors:** Mirella Stańdo, Paweł Piatek, Magdalena Namiecinska, Przemysław Lewkowicz, Natalia Lewkowicz

**Affiliations:** 1Department of Periodontology and Oral Diseases, Medical University of Lodz, 90-419 Lodz, Poland; mirella.stando@umed.lodz.pl; 2Department of Neurology, Medical University of Lodz, 90-419 Lodz, Poland; pawel.piatek@umed.lodz.pl (P.P.); magdalena.namiecinska@umed.lodz.pl (M.N.); przemyslaw.lewkowicz@umed.lodz.pl (P.L.)

**Keywords:** periodontitis, non-surgical treatment, eicosapentaenoic acid, docosahexaenoic acid, salivary cytokines, salivary chemokines, salivary growth factors

## Abstract

Periodontitis is a chronic multifactorial inflammatory disease that leads to the loss of supportive tissues around the teeth with gradual deterioration of masticatory function and esthetics, resulting eventually in the decrease of the life quality. Host immune response triggered by bacterial biofilm is responsible for the chronic periodontal inflammation and ongoing tissue loss. Omega-3 polyunsaturated fatty acids (PUFA) such as eicosapentaenoic acid (EPA) and docosahexaenoic acid (DHA) have anti-inflammatory properties, thus may be used for the treatment of chronic inflammatory diseases. In this study, we aimed to evaluate the effect of dietary supplementation with omega-3 PUFA in the patients with stage III and IV periodontitis. Thirty otherwise healthy patients were treated with scaling and root planning (SRP). In the test group (*n* = 16), patients were additionally supplemented with 2.6 g of EPA and 1.8 g of DHA. In the control group (*n* = 14), patients received only SRP. Periodontal examination was performed at baseline and three months following initial therapy. Salivary samples were taken twice at baseline and at the end of the experiment. We found that there was a statistically significant reduction in the bleeding on probing (BOP) and improvement of clinical attachment loss (CAL) at three months in the test group compared to the control group. Moreover, a statistically significant higher percentage of closed pockets (probing depth ≤ 4 mm without BOP) was achieved in the test group vs. control group after three months of treatment. Accordingly, the levels of pro-inflammatory cytokines/chemokines interleukin (IL)-8 and IL-17 were markedly lower, while the level of anti-inflammatory IL-10 was significantly higher in the salivary samples of the patients supplemented with omega-3 PUFA at three months in comparison to the patients treated with SRP alone. Our findings demonstrate that dietary intervention with high-dose of omega-3 PUFA during non-surgical therapy may have potential benefits in the management of periodontitis.

## 1. Introduction

Periodontitis is a highly prevalent oral disease in humans, affecting nearly 50% of the population worldwide [1]. The Global Burden of Disease 2015 study conducted by Kassebaum et al. revealed that the severe form of periodontitis affects 11.2% of the world’s population and it is the sixth most common human disease [2]. Periodontitis is a multifactorial disease individually accelerated or decelerated by different factors. One of them, a bacterial biofilm, leads to dysbiosis and an increase in Gram-negative bacteria, mainly *Porphyromonas gingivalis*, *Aggregatibacter actinomycetemcomitans*, *Tannerella forsythia*, *Prevotella intermedia*, *Treponema denticola* and *Fusobacterium nucleatum ssp*., in gingival sulcus [3]. This results in the immune response activation and clinical signs of periodontal tissue inflammation. Bacterial burden elicits an increase in lipopolysaccharide, resulting in an increase in pro-inflammatory mediators such as interleukin (IL)-1 β, IL-6 and tumor necrosis factor (TNF)-α [4]. This inflammation, in turn, provokes osteoclastogenesis modifying the levels of receptor activator on nuclear factor-kappaB ligand (RANKL) and/or osteoprotegerin (OPG) [5]. Periodontal inflammation eventually proceeds to the loss of tooth-supporting structures, including connective tissue attachment and alveolar bone. Tissue destruction appears as a result of the interaction between dental plaque bacteria and the immune inflammatory response; however, most of the tissue damage is caused by the host response to infection, not directly by the infectious agents [6,7].

Nowadays, knowing a lot about the mechanisms of inflammation in periodontitis, host modulation therapy (HMT) seems to be an adequate concept for the treatment of periodontal diseases. The main assumption of this therapy is to reduce bystander tissue destruction, to ensure rapid resolution of inflammation or even to promote regeneration of the periodontal tissues by modifying or downregulating the destructive aspects of the host response as well as by upregulating the protective or regenerative responses [8]. The concept of that treatment is to enrich standard periodontal therapies (non-surgical or surgical approach) with HMT delivered systemically or locally. As a result, wound healing and periodontal stability without impairing normal defense mechanisms or inducing inflammation can be achieved [9].

Recently, an increased interest in modulating standard non-surgical treatment of periodontitis using omega-3 polyunsaturated fatty acids (PUFA) as an adjunctive therapy has been observed [10,11,12,13,14,15]. Omega-3 PUFA, including docosahexaenoic acid (DHA) and eicosapentaenoic acid (EPA), have been shown to have a wide range of effects, including anti-inflammatory, immunoregulatory and antioxidant-enhancing properties [15,16,17]. It turns out that omega-3 PUFA have therapeutic and protective qualities in managing of various inflammatory diseases, such as rheumatoid arthritis, ulcerative colitis, asthma, atherosclerosis, cardiovascular disease and periodontitis [18].

In previous human studies, low-dose omega-3 PUFA was administered alone or in combination with acetylsalicylic acid in the treatment of periodontitis with different outcome [10,11,12,13,14,15], leading to the conclusion that the effect of omega-3 PUFA could be cumulative and time-dependent. Taking into account previous studies, we assumed that the duration of omega-3 PUFA supplementation in patients with chronic inflammatory disease such as periodontitis should be not shorter than three months with an adequately high dose of omega-3.

The goal of the present study was to assess the effect of high-dose omega-3 PUFA EPA and DHA on the clinical outcome of non-surgical treatment of the patients with generalized stage III and IV periodontitis. We presumed that dietary supplementation with high-dose EPA and DHA would have the potential to induce a measurable clinical outcome as a result of reduction of inflammation and minimizing tissue damage mediated by anti-inflammatory effect of omega-3 PUFA. To address this issue, we designed a randomized clinical trial in which EPA and DHA were supplemented in adjunction to the standard periodontal therapy, scaling and root planning (SRP). Clinical outcomes of active versus control therapies were measured in addition to the quantifications of salivary cytokines, chemokines and growth factors.

## 2. Materials and Methods

The study was designed as a parallel-arm, randomized clinical trial. A three-month trial period was set: (i) baseline; (ii) three weeks (oral hygiene reinforcement); and (iii) three months (short-term outcome assessment). The study was conducted in accordance with the Declaration of Helsinki, and the protocol was approved by Medical University of Lodz Bioethics Committee (Reference number RNN/251/17/KE) and registered at clinicaltrials.gov (NCT04477395). Patients were recruited at the Department of Periodontology and Oral Diseases, Medical University of Lodz. After the nature of the study was clearly explained to participants, they signed the consent form for participation.

### 2.1. Eligibility and Exclusion Criteria

Forty patients aged 22–70 years (21 females, 19 males, mean age 48.4 ± 10.59) with generalized stage III and IV periodontitis were selected for this study. The criteria of the Classification of Periodontal and Peri-Implant Diseases and Conditions 2017 [19] were used for periodontitis staging. Stage III periodontitis was diagnosed when the following criteria were met: interproximal clinical attachment level (CAL) ≥ 5 mm in at least two non-adjacent teeth, probing depth (PD) ≥ 6 mm and radiographic bone loss beyond one-third coronal part of the root. Stage IV periodontitis was diagnosed when, in addition to the clinical parameters of stage III, tooth loss ≥ 5 teeth with an impairment of occlusion and masticatory function were found. The inclusion criteria were the following: at least 18 scorable teeth (not including third molars), ≥ 4 teeth with PD ≥ 6 mm, CAL ≥ 5 mm, radiographic evidence of bone loss more than one-third of the root length and no periodontal treatment performed within last six months. The exclusion criteria were the following: smoking, history of diabetes or chronic inflammatory disease, any diseases that compromise wound healing, history of radio- or chemotherapy, history of nonsteroidal anti-inflammatory drug (NSAIDs) intake >3 days and use of antibiotics or corticosteroids three months prior to the study.

### 2.2. Study Design

All assessments and treatments were carried out at the Department of Periodontology and Oral Diseases, Medical University of Lodz over a two-year period (from October 2017 to June 2020). At the beginning, patients filled in the questionnaire about their general health and diet, followed by the assessment of clinical and radiological periodontal status. Following screening, patients were invited to participate in the study. All subjects were randomly assigned to one of the two treatment groups by using a website (www.randomization.com). The test group consisted of 20 patients (10 males, 10 females, mean age 47.5 ± 9.63) and the control group consisted of 20 patients (9 males, 11 females, mean age 49.3 ± 12.80).

All participants received non-surgical periodontal therapy. SRP was performed using ultrasonic (Piezon Master 700, EMS) and hand (Mini Five Gracey Curette, Hu-Friedy) instrumentation under a local anesthesia if necessary. The number of SRP sessions varied from 2 to 4 depending on the extent and severity of periodontitis. All patients were instructed about etiology and consequences of periodontal disease and received individual oral hygiene instructions. SRP was repeated at the follow-up visit at three months only at the sites with PD ≥ 4 mm positive for bleeding on probing (BOP). Furthermore, to ensure adequate oral hygiene performance, patients appeared to the recall visit at Week 3 after completion of initial SRP sessions. Oral hygiene was reevaluated by dental plaque disclosing (HurriView II, Beutlich LP, Bunnell, FL, USA), and toothbrushing technique and the use of interdental cleaning devices (interdental toothbrushes or dental floss) were checked and reinforced.

In the control group, patients received SRP only. In the test group, SRP was supplemented with the dietary fish oil (FO) rich in omega-3 PUFA EPA and DHA for three months. Fish oil (BioMarine Medical, liquid, UPRP patent # P.416768), derived from *Centroscymnus crepitater*, *Etmopterus granulosus*, *Deania colceai*, *Centrophorus scalpratus*, *Sardinops sagax*, *Scomber scombrus* and *Gadus morhua* species, was administered twice a day at a dose of 10 mL. Daily dose of 20 mL provided 2.6 g of EPA, 1.8 g of DHA, 1.4 g of alkylglycerols, 1.4 g of squalene, 240 µg of vitamin A and 2 µg of vitamin D3. Patients were required to fill in a diet diary documenting the daily consumption of fish oil and complaints about its consumption. The compliance with fish oil intake was monitored by calling the patients every four weeks during the medication period to check the bottles back for any possible remaining oil. Then, patients received required bottles of the fish oil for subsequent four weeks.

Periodontal charting was performed by a periodontist (N.L.) blinded to the study allocation of participants. All treatments were performed by M.S. who knows the study allocation of the patients. Both charting and treatments were done under magnification (PeriOptix TTL Ready-Made 2.7× or 3.1× loups).

### 2.3. Clinical Assessment

Clinical parameters were evaluated at two time points: at baseline and three months. The following clinical parameters were assessed: full mouth plaque index (FMPI), BOP, PD and gingival recession (REC). CAL was calculated as a sum of PD and REC at respective sites. All measurements were carried out using an UNC-15 periodontal probe (Hu-Friedy) and recorded at six sites (buccal, lingual, mesio-lingual, mesio-buccal, disto-lingual and disto-buccal) for each tooth with exclusion of the third molars. A trained and calibrated examiner (N.L.) performed all assessments at baseline and at follow-up. The study examiner (N.L.) participated in a calibration exercise and the standard error of measurement was calculated. The examiner reliability was high with agreement in assessment on all clinical parameters of above 80% (ICC > 0.80).

### 2.4. Saliva Sampling

Saliva collection was completed before clinical periodontal measurements and any periodontal intervention. Patients were advised not to chew gum, eat or drink anything except water 1 h before sampling. Salivary samples were taken from all individuals at baseline and after three months. A total of 5 mL of unstimulated saliva was collected in an empty 50-mL Falcon tube before clinical measurements. Patients were asked to lean their heads forward and kept their mouths slightly open with minimal head movement to allow passive drainage of saliva into the test tube. The samples were centrifuged at 2620 rpm (10,000 *g*) for 10 min at 4 °C degrees. Subsequently, the supernatants were collected at 1.5 mL Eppendorf tubes and frozen at −80 °C.

### 2.5. Multiple Profiling Chemokine/Cytokine Assays

Concentrations of fifty-four cytokines, chemokines and growth factors in saliva supernatants were measured using Bio-Plex Pro™ Human Chemokine Assays (Bio-Rad Laboratories) and Bio-Plex Pro^TM^ Human Cytokine 27-plex Assay.

Standards and samples were diluted (1:4) in sample diluent and transferred to the plate containing magnetic beads for 1 h at RT. Next, the plate was washed (3×) and detection antibody was added for 30 min on a shaker (850 rpm) at RT. After that, the plate was washed (3×) and streptavidin–PE solution was added for 10 min. Subsequently, the plate was washed (3×) and samples were re-suspended in 125 µL of assay buffer and analyzed within 15 min. All samples were analyzed at the same time in duplicates. All reagents and technology were provided by Bio-Rad Laboratories (Bio-Plex 200).

### 2.6. Statistical Analysis

Sample size was estimated based on the primary outcome parameter (mean PD) from a study employing 300 mg of n-3 PUFA during non-surgical periodontal treatment [12]. The study was powered at 80% to detect a mean PD difference of 0.6 mm (SD ± 0.53) between test and control groups at three months. The minimum required sample size was calculated to be 14 patients for each group; to compensate for potential dropouts, 20 patients were recruited for each group.

The statistical unit was the patient, and all sites with PD ≥ 4 mm at baseline were considered for the statistical analyses. The primary outcome variable was the percent of closed pockets (PD ≤ 4 mm and BOP) at three months in relation to baseline. Secondary variables were average changes in PD, CAL, REC, BOP, FMPI, number of sites with PD ≥ 5 mm and concentrations of cytokines/chemokines/growth factors. Additionally, for clinical parameters, changes (Δ) from baseline to three months were calculated at subject level.

Following calculation of the mean ± SD of each parameter for both groups, statistical comparison of differences within the groups at two time points were determined by the Paired *t*-test, and the comparisons between the test and control groups were performed using the unpaired Students *t*-test. The verification of normal distribution and analysis of variances were made using the Kolmogorov–Smirnov test and the Fisher’s test. Changes (Δ) from baseline to three months between the groups were analyzed using Mann–Whitney U test. *p* < 0.05 was considered as the significant difference.

## 3. Results

### 3.1. Dietary Supplementation with Omega-3 PUFA Resulted in the Improvement of CAL and BOP

Forty patients meeting inclusion criteria were included in the study. Ten subjects were lost during follow-up because of the following reasons: antibiotic or NSAIDs intake for other medical reasons (*n* = 4), severe acute respiratory syndrome coronavirus 2 (SARS-CoV-2) pandemic-related lockdown (*n* = 5) and moving to another country (*n* = 1). Thus, per protocol, analyses included only thirty patients aged 30–70 years (mean age: 49.0 ± 10.59 years old) (Figure 1).

According to the new classification of periodontal diseases 2017 [19], all the patients presented generalized periodontitis stage III or IV. The demographic distribution revealed statistical difference regarding age between the test and control groups (Table 1).

No adverse events or anomalies were observed in oral soft and hard tissue examinations. In the intervention group, six subjects reported nausea and irritating fish-scented halitosis as adverse effects that were not strong enough to stop the treatment regimen. No other adverse events were reported.

Baseline periodontal parameters did not differ between the groups, except for FMPI (Table 2). Throughout the study, plaque accumulation was markedly reduced in both groups; however, FMPI remained significantly higher in the control group at three months. BOP scores showed higher reduction at three months in the patients receiving omega-3 PUFA compared to the control group (*p* < 0.05). In both groups, the number of pockets and PD reduced after three months compared to the baseline but there were no statistically significant differences between the groups (*p* > 0.05). There was a statistically significant improvement of CAL and REC mean values in the test group vs. control group after three months following treatment. Moreover, a statistically significant higher percentage of closed pockets (PD ≤ 4 mm without BOP) was achieved in the test group vs. control group at three months (Table 2).

Taken together, the patients receiving omega-3 PUFA demonstrated better improvement in resolution of inflammation (greater reduction of BOP) and higher gain of CAL compared to the patients treated with SRP alone.

### 3.2. Omega-3 PUFA Affected the Levels of Key Salivary Cytokines, Chemokines and Growth Factors

To address whether clinical improvement in the test group in comparison with the control group was accompanied by the appropriate changes in the pro- and anti-inflammatory mediators, we compared salivary levels of fifty-four cytokines/chemokines/growth factors using multiELISA. At baseline, no differences in the mean concentrations of cytokines/chemokines/growth factors between the test and control groups were detected. Conversely, at three months, we found that mean concentrations of pro-inflammatory IL-8 and IL-17 were markedly lower, and the concentration of anti-inflammatory IL-10 was significantly higher in the salivary samples of the patients who received omega-3 PUFA in comparison to the patients treated with SRP alone (Table 3). Moreover, IL-12 level was increased at three months in both tested groups in comparison to the baseline, suggesting reestablishing the balance between T helper (Th) 1- and Th2-type immune response towards Th1. This increase was significantly higher in the test group (*p* = 0.03). We also detected different pattern of chemokine release in the patients receiving FO in comparison to the control group. At three months, we detected significant increase of chemokine (C-C motif) ligand (CCL) 4, CCL5, CCL15, CCL25 and chemokine (C-X3-C motif) ligand (CX3CL) 1 and decrease of CCL21, CCL26, CCL27, chemokine (C-X-C motif) ligand (CXCL) 1, CXCL10 and CXCL16 in the test group (Table 3). These changes may reflect cell signaling during tissue healing and repair [20]. Further, we found a significantly higher concentration of fibroblast growth factor (FGF) 2 in saliva in the test group vs. control group at three months. This was a result of a statistically insignificant increase of FGF2 in the test group (*p* = 0.07) and its unchanged level in the control group (*p* = 0.49) after treatment (Table 3). FGF2 is a multipotent factor responsible for angiogenesis, keratinocyte organization and wound healing processes. Finally, we detected different tendency in granulocyte-colony stimulating factor (G-CSF) concentrations between the groups. In the test group, G-CSF decreased at three months (*p* = 0.02), whereas, in the control group, it increased, but the change was statistically insignificant (*p* = 0.09) (Table 3).

Collectively, we demonstrated regulatory effects of high-dose omega-3 PUFA on mediators of periodontal inflammation and healing. This shift of immune response promoted better resolution of inflammation detected clinically as reduced rates of BOP and improved healing detected clinically as CAL improvement and better rates of closed periodontal pockets.

## 4. Discussion

The present study showed that supplementation with high-dose omega-3 PUFA as an adjunct to non-surgical treatment of periodontitis can be helpful in management of periodontal disease. It is well recognized that mechanical debridement is a key element in the treatment of periodontal inflammation, with SRP being a gold standard of non-surgical treatment. As expected, the improvement of clinical parameters was demonstrated at three months in comparison with baseline in both the study and control groups. We found, however, a statistically significant better improvement in BOP and CAL at three months in the patients receiving omega-3 PUFA in comparison with SRP alone. These parameters are sensitive indicators of resolution of inflammation and tissue healing that both are crucial in preserving healthy and stable dentition for a long period of time in patients with periodontitis. We confirmed that EPA and DHA can soothe the ongoing inflammation in periodontitis and can be used as HMT non-surgical treatment.

Our results are in accordance with the study of Deore et al. demonstrating that dietary supplementation of omega-3 PUFA with non-surgical periodontal treatment had significant effect on clinical parameters such as gingival index (GI), BOP, PD and CAL, but no effect on serum C-reactive protein (CRP) level [12]. In contrast, other studies showed that using omega-3 PUFA during non-surgical periodontal treatment had no effect on clinical parameters [10,13]. These differences can result from much lower doses of omega-3 PUFA than used in our study. Keskiner et al. showed that daily dietary supplementation with low-dose omega-3 PUFA (6.25 mg EPA and 19.19 mg DHA) for six months may reduce salivary TNF-α, however no significant impact on clinical parameters was gained [10]. It was concluded that the effect of omega-3 PUFA could be cumulative and time-dependent. In another study, the effect of daily consumption of 540 mg EPA and 360 mg DHA for 12 months was analyzed. A significant increase of EPA level and decrease of the arachidonic acid/EPA serum ratio (mean levels of 7:1) was achieved [13]. Earlier studies suggested that optimal omega-6/omega-3 ratio should be approximately 4:1–5:1, and not exceeding 10:1, to maintain health [21,22,23]. This suggests that higher intake of omega-3 PUFA for a longer period can result in better outcome. Therefore, in our study, we decided to use high-dose omega-3 PUFA (2.6 g of EPA and 1.8 g of DHA). This is in line with the studies demonstrating beneficial effects of omega-3 PUFA supplementation in other chronic inflammatory diseases such as rheumatoid arthritis (7.1 g of omega-3 PUFA per day) or Crohn’s disease (2.7 g of omega-3 PUFA per day) [24]. To our surprise, the clinical effects of 2.6 g of EPA and 1.8 g of DHA in our study were clearly detectable at three months, thus dietary intervention with high-dose omega-3 PUFA seems to provide rapid anti-inflammatory effect detectable at the clinical level. Moreover, omega-6 and omega-3 PUFA are essential because of the inability of humans, and all mammals alike, to synthesize them. Therefore, they must be obtained from properly balanced diet. It has been estimated that current Western diet is deficient in omega-3 PUFA with a ratio of omega-6 to omega-3 of 15–20/1. Today, industrialized societies can be characterized by an increased intake of saturated fat, omega-6 fatty acids and trans fatty acids and a decreased consumption of omega-3 fatty acids [25,26].

The effects of omega-3 PUFA were also tested in the patients with gingivitis. Similar to the results of our study, the decrease of BOP was demonstrated in the patients that received omega-3 PUFA in addition to the scaling and oral hygiene instructions versus control group, thus anti-inflammatory properties of EPA and DHA were also clinically detectable at the less advanced stages of periodontal inflammation [27,28].

Some studies employed acetylsalicylic acid together with omega-3 PUFA [10,11,14,15]. Simultaneous administration of omega-3 PUFA and low-dose aspirin resulted in a synergistic interaction of both compounds in resolution of inflammation. Elwakeel and Hazaa revealed in the patients with diabetes mellitus a significant improvement in clinical parameters plaque index (PI), GI, PD and CAL in the test group versus control group, and a highly significant reduction in the IL-1β level in the gingivocrevicular fluid (GCF) in the experimental group [11]. Another study demonstrated a significant reduction in PD and CAL in the group receiving omega-3 PUFA plus aspirin during non-surgical periodontal treatment compared to the control group, with significantly reduced levels of salivary RANKL and matrix metalloproteinase (MMP) 8 in the test group [29]. One study tried to determine whether low-dose of 81 mg of acetylsalicylic acid in combination with a total daily dose of approximately 2 g of DHA without mechanical debridement would be beneficial in the treatment of periodontitis versus control group where only 81 mg of aspirin + corn/soy oil capsules was administered. In the intervention group with DHA, decreases in PD and GI and no changes in PI and BOP were demonstrated with the reduced levels of CRP and IL1-β in GCF. Moreover, when acetylsalicylic acid was used alone, no beneficial effect on periodontal inflammation was noted [15]. Although the results of the above-mentioned studies are favorable, it should be kept in mind that providing patients with low-dose aspirin as an adjunct to non-surgical treatment might be questionable. Firstly, long-term intake of aspirin is mainly recommended in patients as a primary and secondary prevention of cardiovascular events, where possible adverse effects are less dangerous than the progression of cardiovascular disease [30]. Secondly, acetylsalicylic acid taken for a long period of time inhibits constitutive cyclooxygenase-1, which is involved in maintaining physiological homeostasis [27].

Another goal of our study was to determine the effect of omega-3 PUFA on the inflammatory mediators released during periodontal inflammation. The levels of cytokines/chemokines/growth factors were analyzed in the salivary samples at three months. Saliva is a good medium for detection of soluble mediators released during periodontal inflammation into gingivocrevicular fluid and subsequently to the saliva. We decided to investigate salivary samples because of the simplicity of the methodology in comparison with GCF collection and analysis [14,28]. Saliva as a non-invasive diagnostic fluid was previously used for determining the inflammatory status of periodontal patients with good results [31,32]. It is generally agreed that non-surgical periodontal treatment can result in downregulation of pro-inflammatory cytokines associated with bone connective tissue and bone metabolism. The majority of studies demonstrated significant reduction of salivary concentrations of Il-1β, IL-6, IL-8, TNF-α and GM-CSF, as well as increases in IL-4 and IL-10 [33,34,35,36,37].

However, some studies failed to show the same pattern pointing to quite high variability of salivary cytokine levels in the patients with periodontitis [38,39]. In our study at three months after SRP, the reduction of IL-1 β was statistically significant in the control group, while the reductions of IL-2, IL-8, IL-17, TNF, CCL21, CCL26, CCL27, CXCL1, CXCL10, CXCL16 and G-CSF were significant in the test group.

When comparing the effect of fish oil on the inflammatory mediators, we found that the levels of proinflammatory IL-8 and IL-17 were markedly lower, and the level of anti-inflammatory IL-10 was significantly higher in the salivary samples of the patients that received omega-3 PUFA in comparison with the patients treated SRP alone. Both IL-8 and IL-17 are known for neutrophil recruitment, and an important role of IL-17 signaling in tissue damage during periodontal inflammation was recently described [40]. Resolvin D2 (RvD2), a product of DHA metabolism, was shown to prevent alveolar bone loss in *Porphyromonas gingivalis*-induced experimental periodontitis by inhibiting Th1/Th17 polarization [41]. Moreover, in the mouse model of periodontitis, it was demonstrated that IL-10 plays a protective role by dampening an excessive IL-17–mediated inflammatory response majorly through innate immune cells [42]. In line with these findings, we previously reported that IL-10 may induce a population of IL-10-producing neutrophils in periodontitis [43].

We also detected an increased salivary concentration of IL-12 in the test group after treatment. IL-12 induces Th1 cell differentiation from naive T cells and their subsequent IFN-γ production. Previous studies showed that IL-12 and IFN-γ did not have a major effect on the pathogenesis of infection-stimulated bone resorption in vivo [44], and short-term non-surgical therapy resulted in a significant improvement in periodontal indices and a marked increase in the salivary IL-12 levels [45]. However, an osteolytic role of IL-12 in the pathogenesis of periodontitis was also suggested [46]. Interestingly, IFN-γ, a Th1-type cytokine that was increased in the saliva of the test group seems to act antagonistically on Th17 differentiation [47] and osteoclastogenesis [48].

In this study, we also found that an intake of fish oil rich in omega-3 PUFA slightly promoted an increase of salivary concentrations of FGF2 that was previously shown to promote bone formation through accelerating the differentiation of osteoprogenitor cells and to stimulate proliferation and migration of periodontal ligament cells [49].

Host modulatory therapy is one of the main focuses of interest for many of the investigators and currently considered a promising treatment approach. Many medications were proposed in this context such as non-steroidal anti-inflammatory drugs, tetracyclines or bisphosphonates which may be delivered locally or systemically [7,9,50]. However, those pharmacological substances can be used only to a limited extent and have undesirable effects. Thereby, our attention was focused on safer, naturally derived PUFA EPA and DHA. Although the studies analyzing omega-3 PUFA in the treatment of periodontitis are limited, and the study design differed in the aspects such as EPA and DHA dose, duration of study, number of participants and inclusion criteria, a general conclusion can be drawn that this approach provides favorable clinical outcomes. We need to emphasize that non-surgical periodontal treatment is required for the patients in terms of cooperation and proper regimen. In particular, adequate daily oral hygiene and controlling the risk factors, such as smoking habit or diabetes, may be crucial in the long-term maintenance of disease remission. Keeping in mind the systemic effects of untreated periodontal inflammation on general health, introduction of the natural and safe dietary supplementation during treatment of periodontitis may have additional beneficial effects. In this context, omega-3 PUFA seem to be particularly beneficial in the patients with hypercholesterolemia/cardiovascular diseases and periodontitis [51].

Our study has a few limitations. One of them is the baseline difference between the test and control groups in terms of age and FMPI. This was a result of a relatively high dropout rate, mainly due to SARS-CoV-2 epidemic-related lockdown. When we took into account the baseline data of all forty patients that were recruited, no differences between the groups in the age and FMPI were noted. Although both groups at baseline have comparable advancement of periodontitis, we need to bear in mind that age and plaque accumulation are the risk factors of periodontitis and may lead to the worse treatment response [52]. However, it has been suggested that the increased level of periodontal destruction observed with aging is the result of cumulative damage rather than a result of its increased rates [53]. Thus, aging is not a risk factor per se. There are very limited data on periodontal healing in older individuals showing both delayed [54] and normal healing after periodontal treatment of people with moderate-to-advanced forms of periodontitis [55]. In general, aging is not perceived as a negative predictor for unfavorable treatment response and risk factor for disease progression [19]. Another limitation of the study is the use of fish-derived oil mixture instead of isolated EPA and DHA. Natural fish oil was chosen to mimic dietary conditions where a range of essential oils is consumed. However, we cannot rule out that other components of the fish oil affected the outcome of our study. Specifically, squalene and alkylglycerols present in the fish oil may affect inflammatory immune response activating Th1-type IL-12 and IFN-γ cytokine production and increasing total antioxidant status of serum [56,57]. Because of the choice of fish oil as a source of EPA and DHA, no placebo was employed in our study and patients were aware of the treatment allocation. In our experience, the taste of fish oil cannot be masked, even when it is taken in capsules (as a result of belching). Future research with a larger sample size and long-term observation are warranted to validate the usage of omega-3 PUFA as an adjunctive dietary therapy option to treat periodontitis.

## 5. Conclusions

In conclusion, the results obtained in this study suggest that daily dietary supplementation with high-dose omega-3 PUFA can be used as an adjunct to non-surgical treatment of periodontist as host modulatory therapy. No adverse events were recorded during omega-3 PUFA treatment. Omega-3 PUFA had favorable effects on resolution of inflammation and tissue regeneration in the patients with periodontitis. Our research showed significant improvement of clinical parameters accompanied by the reduction of salivary levels of pro-inflammatory cytokines/chemokines in favor of higher level of anti-inflammatory IL-10 and pro-regenerative FGF2.

## 6. Patents

Biomarine Medical fish oil composition is restricted by patent No. UPRP P.416768.

## Figures and Tables

**Figure 1 nutrients-12-02614-f001:**
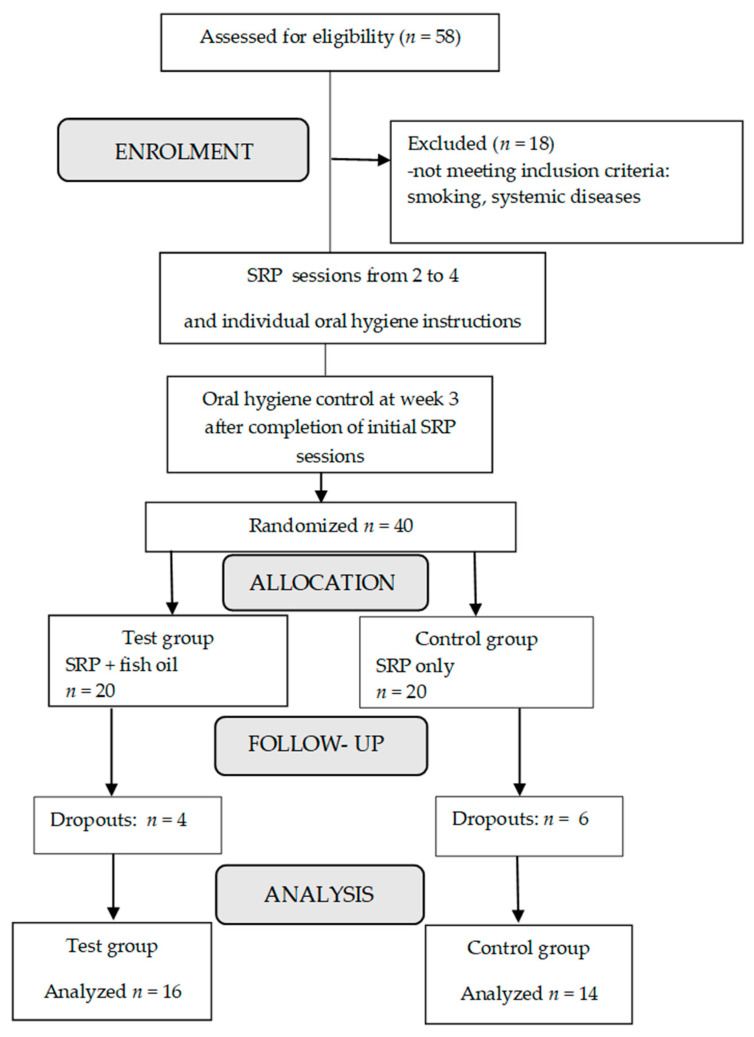
Flowchart of the study. SRP, scaling and root planning.

**Table 1 nutrients-12-02614-t001:** Demographic characteristics of the groups.

	SRP Plus Fish Oil	SRP Alone	*p* Value
Age (years; mean ± SD)	45 ± 8	54 ± 11	0.005
Sex (% of females)	50	43	0.349

**Table 2 nutrients-12-02614-t002:** Mean scores of clinical parameters with their Δ at baseline and three months in the test (SRP plus fish oil) and control (SRP alone) groups. Data are mean ± SD.

Variables	SRP Plus Fish Oil *n* = 16	SRP Alone *n* = 14	Inter-Group Comparisons *p* Value
Number of PD ≥ 4 mm			
Baseline	55 ± 29	52 ± 17	0.337
3 months	32 ± 21	32 ± 17	0.461
Δ baseline–3 months	24 ± 18	20 ± 7	0.222
PD (mm)			
Baseline	5.0 ± 0.5	5.1 ± 0.8	0.426
3 months	3.7 ± 0.7	4.0 ± 0.7	0.461
Δ baseline–3 months	1.3 ± 0.7	1.1 ± 0.4	0.124
REC (mm)			
Baseline	0.8 ± 0.6	1.1 ± 0.9	0.121
3 months	0.8 ± 0.5	1.3 ± 0.7	0.023
Δ baseline–3 months	0.1 ± 0.3	0.2 ± 0.4	0.115
CAL (mm)			
Baseline	5.8 ± 0.8	6.1 ± 1.1	0.144
3 months	4.4 ± 1.1	5.3 ± 1.0	0.017
Δ baseline–3 months	1.3 ± 0.7	0.9 ± 0.4	0.021
BOP (%)			
Baseline	28 ± 16	36 ± 19	0.069
3 months	14 ± 6	21 ± 7	0.004
Δ baseline–3 months	13 ± 15	16 ± 21	0.335
FMPI (%)			
Baseline	35 ± 21	49 ± 19	0.031
3 months	17 ± 9	34 ± 16	0.0004
Δ baseline–3 months	18 ± 19	15 ± 23	0.345
Closed pockets (%) with PD ≤4 mm and no BOP			
Baseline	0	0	-
3 months	58 ± 17	49 ± 11	0.042
Δ baseline–3 months	58 ± 17	49 ± 11	0.042

Abbreviations: probing depth (PD), gingival recession (REC), clinical attachment level (CAL), bleeding on probing (BOP), full-mouth plaque index (FMPI).

**Table 3 nutrients-12-02614-t003:** The profile of salivary cytokines, chemokines and growth factors at baseline and three months. Data are presented in pg/mL as means ± SD.

	SRP Plus Fish Oil		SRP Alone	
	Baseline	3 Months	Baseline	3 Months
Cytokines				
IL-1β	223 ± 147	206 ± 144	330 ± 273	199 ± 176 ^†^
IL-1RA	5764 ± 2186	6804 ± 3095	5555 ± 3851	5671 ± 3235
IL-2	21 ± 12	13 ± 4 ^†^	15 ± 5	15 ± 8
IL-4	22 ± 12	18 ± 5	17 ± 10	15 ± 4
IL-5	7.6 ± 5.7	9.4 ± 5.8	5.4 ± 4.8	7.1 ± 5.5
IL-6	110 ± 82	139 ± 115	193 ± 150	133 ± 137
IL-7	31 ± 20	26 ± 15	20 ± 10	48 ± 13
IL-9	11 ± 6	14 ± 8	10 ± 4	12 ± 6
IL-10	149 ± 51	211 ± 65 *^,†^	133 ± 34	129 ± 44
IL-12	1.7 ± 1.3	3.2 ± 1.8 *,^†^	0.9 ± 0.5	1.6 ± 0.9 ^†^
IL-13	0.5 ± 0.2	0.5 ± 0.4	0.5 ± 0.2	0.5 ± 0.6
IL-15	nd	44 ± 37	nd	49 ± 13
IL-16	2600 ± 2680	1228 ± 872	1163 ± 754	1556 ± 1596
IL-17	17 ± 8	8 ± 3 *^,†^	12 ± 3	11 ± 2
IFN-γ	38 ± 17	46 ± 29	46 ± 21	38 ± 16
MIF	677,488 ± 645,516	445,004 ± 302,624	345,344 ± 289,914	280,090 ± 287,976
TNF-α	54 ± 29	41 ± 26 ^†^	37 ± 13	33 ± 10
Chemokines				
CCL1/I-309	38 ± 12	34 ± 9	29 ± 10	27 ± 9
CCL2/MCP-1	1205 ± 870	1515 ± 1795	1194 ± 689	1642 ± 1579
CCL3/MIP-1 α	33 ± 31	47 ± 63	16 ± 10	17 ± 16
CCL4/MIP-1 β	8.7 ± 5.8	18 ± 16 ^†^	7.3 ± 1.7	9.4 ± 5.1
CCL5/RANTES	4.8 ± 2.3	7.2 ± 1.6 *^,†^	4.2 ± 3.5	4.2 ± 2.3
CCL7/MCP-3	33 ± 27	28 ± 23	24 ± 10	26 ± 11
CCL8/MCP-2	3.9 ± 5.0	2.4 ± 1.5	1.6 ± 1.0	2.1 ± 1.4
CCL11/Eotaxin	28 ± 8	28 ± 8	24 ± 7	23 ± 5
CCL13/MCP-4	98 ± 63	96 ± 68	57 ± 34	70 ± 42
CCL15/ MIP-1delta	127 ± 87	268 ± 196 ^†^	271 ± 319	181 ± 166
CCL17/TARC	7.2 ± 7.4	2.3 ± 2.9	7.3 ± 4.6	2.8 ± 3.1
CCL19/MIP-3 β	124 ± 51	119 ± 52	99 ± 34	153 ± 101
CCL20/MIP-3 α	6.7 ± 6.0	10 ± 20	3.8 ± 3.2	3.6 ± 3.4
CCL21/6Ckine	2329 ± 1891	1297 ± 746 ^†^	1370 ± 861	1750 ± 1119
CCL22/MDC	32 ± 12	33 ± 16 *	25 ± 10	21 ± 7
CCL23/MPIF-1	30 ± 170	26 ± 14	25 ± 10	17 ± 7
CCL24/Eotaxin-2	25 ± 31	20 ± 8	18 ± 6	23 ± 5 ^†^
CCL25/TECK	827 ± 316	843 ± 408*^,†^	533 ± 126	523 ± 142
CCL26/Eotaxin-3	24 ± 10	15 ± 7 ^†^	13 ± 5	12 ± 4
CCL27/CTACK	18 ± 12	11 ± 7 ^†^	14 ± 9	12 ± 9
CX3CL1/ Factalkine	1858 ± 1336	2625 ± 1791*^,†^	1422 ± 874	1187 ± 828
CXCL1/Gro-alpha	10,812 ± 11,390	5212 ± 5106 ^†^	3113 ± 1802	3626 ± 3493
CXCL2/Gro-beta	1115 ± 1329	669 ± 705	591 ± 371	439 ± 369
CXCL5/ENA-78	20,513 ± 21,340	19,718 ± 24,877	12,199 ± 14,653	16,399 ± 18,018
CXCL6/GCP-2	163 ± 200	188 ± 339	22 ± 18	50 ± 60
CXCL8/IL-8	20,811 ± 18,099	8580 ± 10,551 *^,†^	41,658 ± 37,582	29,557 ± 28,905
CXCL9/MIG	562 ± 470	625 ± 509	384 ± 416	330 ± 350
CXCL10/IP-10	287 ± 267	173 ± 216†	153 ± 78	140 ± 95
CXCL11/I-TAC	4.0 ± 3.3	3.2 ± 3.2	2.3 ± 2.4	1.7 ± 1.0
CXCL12/SDF-1 α	81 ± 34	84 ± 40	78 ± 23	68 ± 21
CXCL13/BCA-1	5.1 ± 4.8	4.4 ± 5.0	3.1 ± 2.8	4.3 ± 8.4
CXCL16/SCYB16	158 ± 136	88 ± 74 ^†^	100 ± 93	107 ± 147
Growth factors				
FGF2	35 ± 10	40 ± 11 *	29 ± 12	29 ± 9
G-CSF	358 ± 196	222 ± 84 *^,†^	299 ± 142	410 ± 308
GM-CSF	22 ± 9	24 ± 19	20 ± 7	17 ± 5
PDGF-BB	13 ± 6	18 ± 13	10 ± 9	12 ± 9
VEGF	317 ± 228	330 ± 199	293 ± 147	282 ± 219

* Statistically significant differences to SRP alone; ^†^ statistically significant differences between baseline and three months; nd, non detectable; green font—proinflammatory cytokines; pink font—anti-inflammatory cytokines; red font—CXC type chemokines (associated with recruitment of neutrophils and lymphocytes); blue font—CC type chemokines (associated with recruitment of lymphocytes, monocytes, mast cells and eosinophils); black font—growth factors. Abbreviations: interleukin-1 receptor antagonist (IL-1RA), interferon (IFN), macrophage migration inhibitory factor (MIF), granulocyte-macrophage colony-stimulating factor (GM-CSF), platelet-derived growth factor BB (PDGF-BB), vascular endothelial growth factor (VEGF).

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
