# Peer review of "Omega-3 Polyunsaturated Fatty Acids EPA and DHA as an Adjunct to Non-Surgical Treatment of Periodontitis: A Randomized Clinical Trial"

_nutrients, 2020, doi:10.3390/nu12092614_

Round 1
Reviewer 1 Report
This manuscript reports findings from a controlled trial of high dose omega-3 FAs in patients with periodontitis. This is an interesting area. 40 patients began the study but only 30 completed it. Dose pf omega-3 is food as is duration. Omega-3 had some clinical improvements compared to control and also altered cytokines suggesting reduced inflammation. These findings will be of interest.
Comments:
- Control group was medical treatment but no placebo - this is a limitation and should be mentioned.
- Patients were not blinded to treatment allocation - this should be mentioned and included as a limitation.
- Were researchers blinded? Needs a comment.
- Do you have measures of fatty acids in blood?
- Table 4 needs tidying up
Minor comments:
- Line 18 should read "Omega-3 polyunsaturated fatty acids (PUFAs) such as …"
- Throughout. PUFAs omega-3 should read Omega-3 PUFAs
- Line 42. raise of -> increase in
- Line 49. "is eventually proceeding" should read "eventually proceeds"
- Line 62. gained -> achieved
- Line 74. Delete "the"
- Line 115. extend -> extent
- Line 127. 1,4 -> 1.4 (twice)
- Line 245. "medium"??
- Line 280. Should read "The present study showed that .."
- Line 283. Delete "it was"
- Line 304. "can provide with better outcome"??
- Line 392. Delete "it"
Author Response
Reviewer 1
We would like to thank the Reviewer for helpful comments that allowed to improve the manuscript. We addressed all issues raised by the Reviewer. Point-by-point responses are enclosed below.
Comments:
1 and 2. We included both as limitations in the last paragraph of Discussion, lines 406-408: ‘Because of the choice of fish oil as a source of EPA and DHA, no placebo was employed in our study and patients were aware of the treatment allocation. In our experience, the taste of fish oil cannot be masked, even when it is taken in capsules (as a result of belching).’
3. This information was partially mentioned in the Methods and was additionaly supplemented. Line 139-140: ‘Periodontal charting was performed by a periodontist (N.L.) blinded to the study allocation of participants. All the treatments were performed by M.S who knows the study allocation of the patients.’
4. We did not measure the concentrations of PUFAs in blood. We tried to analyze fatty acids using peripheral blood mononuclear cells (PBMCs), to track EPA and DHA binding to the cell wall lipids using gas chromatography. Unfortunately, we failed to achieve reliable results, probably because we used relatively little amount of PBMC isolated from 5 ml of blood.
5. Table 4 (now Table 3) has been modified.
Minor comments:
1 and 2. Checked and corrected throughout the manuscript.
3-8, 10, 11, 13. Corrected exactly as suggested.
9. ‘Medium’ replaced with ‘Mean”.
12. Line 288 corrected into: ‘can result in better outcome’.
Reviewer 2 Report
In the manuscript entitled “Omega-3 polyunsaturated fatty acids EPA and DHA as an adjunct to non-surgical treatment of periodontitis: a randomized clinical trial” the authors tested the effect of high dose omega-3 polyunsaturated fatty acid+scaling and root planning (SRP) on the management of stage III and IV periodontitis. Overall, the study is interesting and potentially publishable. However, it suffers some limitations which the authors do acknowledge. My comments are provided below.
Major comments:
- A significant limitation of the study is the difference in age between the two groups which might have affected the findings. The authors do discuss this point in the last paragraph of the discussion but please discuss further whether healing after SRP is different between different age groups (younger vs older patients)
- Also, the use of fish oil is another problem which they also discuss in the same paragraph. However, according to the authors the daily fish oil dose provided 2.6 g of EPA, 1.8 g of DHA, 1.4 g of alkylglycerols, 1.4 g of squalene, 240 μg of vitamin A and 2 μg of vitamin D3; do any of these non-omega-3 components have any effect on inflammatory mediators and possibly periodontitis? Please discuss
Abstract:
- The authors state “We found that there was a statistically significant reduction in the bleeding on probing (BOP) and improvement of clinical attachment loss (CAL) at 3 months in the test group compared to the control group.” then, the sentence right after states “Moreover, a statistically significant higher percentage of closed pockets with probing depth ≤4 mm and no BOP was achieved in the test group vs. control group after 3 months of treatment” is the effect on BOP repeated in both sentences?
Introduction
- Lines 73-75, the authors state “Taking into account previous studies, we assumed that the duration of omega-3 PUFAs supplementation in the patients with chronic inflammatory disease such as periodontitis should be not shorter than 6 months with and adequately high dose of omega-3.” Why then was the study conducted for 3 months only? Please clarify in the revised version.
Methods
- Please provide criteria of stage III and IV periodontitis under methods
- Lines 123-124, the authors state “In the control group, patients received SRP only. In the test group, SRP was supplemented with the dietary fish oil rich in omega-3 PUFAs EPA and DHA for 6 months” is it 3 or 6 months? Please specify and make it consistent throughout.
- Lines 129-131, the authors sate “For better oil administration control, the subjects had to appear to the Department of Periodontology and Oral Diseases every 4 weeks to receive the required bottles of the fish oil” Did you check patient compliace?
- How was the ICC calculated, was there another rater?
- Line 148, the authors state “Salivary samples were taken from all individuals at baseline and after 3 and 6 months” cytokine/chemokine data are shown for 3 months only. Did you do clinical assessment and cytokine/chemokine measurements at 6 months?
- Line 178, the authors state “comparisons between the test and control groups were performed using the Students t-test” please specify whether it is paired or unpaired
- Table 1, please provide p-value for % of females
Results
- Please merge table 3 with table 2
- Line 237-239, the authors state “BOP scores showed higher reduction at 3 months in the patients receiving omega-3 PUFAs compared to the control group, and results were statistically significant at each time point (p < 0.05)”, there is only one time point (3 months), please delete at each time point or provide data for another time point
- Table 4, too many cytokines/chemokines/growth factors and you only discuss a few of them. Please keep the ones that are relevant and remove any irrelevant ones, the table is so disturbing. I also suggest dividing the table into sections: 1) pro-inflammatory mediators and 2) anti-inflammatory mediators
- For CCL22, FGF2 levels in table 4, the authors considered the significant higher value compared to SRP as an induced effect of omega-3 fatty acid. In fact, these values at 3 months are no different from baseline values of the same group which means that fish oil does not affect them at all. Please revise.
- Also revise G-CSF, was it increased at 3 months in the control group? There are no indicators of significance
Discussion
- Tetracyclines are not non-steroidal anti-inflammatory drugs, please correct
Other minor comments:
- Please revise the article carefully for grammar, spelling and other language-related issues
- Abbreviations should come when the term is first mentioned in full and be consistently used thereafter. Please make sure this is the case for all the abbreviations you used
Author Response
Reviewer 2
We would like to thank the Reviewer for helpful comments that allowed to improve the manuscript. We addressed all issues raised by the Reviewer. Point-by-point responses are enclosed below.
Major comments:
Aging is not perceived as a negative predictor for unfavorable treatment response and risk factor for disease progression. Recent review concluded that studies are needed to address the association of delayed cell proliferation and wound healing with the onset of periodontal diseases and response to treatment (Kanasi E, Ayilavarapu S, Jones J. The aging population: demographics and the biology of aging. Periodontol 2000. 2016 Oct;72(1):13-8. doi: 10.1111/prd.12126.). Appropriate information has been added in the lines 397-400: ‘There is very limited data on periodontal healing in older individuals showing both delayed [55] and normal healing after periodontal treatment of people with moderate-to-advanced forms of periodontitis [56]. In general, aging is not perceived as a negative predictor for unfavorable treatment response and risk factor for disease progression [19].’
We agree that alkylglycerols and squalene present in fish oil may have some biological effect but not vit. A and D, because of the very low dose. Recommended daily intake is 10–20 micrograms of vitamin D and 650-800 micrograms of vitamin A, therefore the amounts of vit. D and A taken in our study had no significant effect on immune response or healing. An appropriate comment has been made in the text, lines 403-405: ’Specifically, squalene and alkylglycerols present in the fish oil may affect inflammatory immune response activating Th1-type IL-12 and IFN-γ cytokine production, and increasing total antioxidant status of serum [57,58].’
Abstract:
‘We found that there was a statistically significant reduction in the bleeding on probing (BOP) and improvement of clinical attachment loss (CAL) at 3 months in the test group compared to the control group.’ – In this sentence BOP is related to all sites measured at baseline (PD≥4 mm).
‘Moreover, a statistically significant higher percentage of closed pockets (probing depth ≤4 mm without BOP) was achieved in the test group vs. control group after 3 months of treatment.’ – In this sentence BOP is related to the sites with PD ≤4 mm. ‘Closed pockets’ means PD ≤4 mm without BOP.
Introduction:
Our study was initially designed as a 6-month RCT and our initial draft of the manuscript was intendent to show the complete data. Unfortunately, SARS-COV-2 pandemic caused significant dropout of the patients that already were close to finish 6-month observation. Therefore, we recently started to recruit new patients to complete the study at 6 months. Because 3-month results were very promising, we decided to publish them. Moreover, screening of salivary cytokines/chemokines/growth factors with multiELISA showed us some new candidates for more in-depth further analyses in the new groups of patients. We corrected all the fragments that mentioned 6-month time point (lines 74-75, 130, 156, 211).
Methods:
The criteria of stage III and IV periodontitis has been provided in the lines 97-103: ,The criteria of the Classification of Periodontal and Peri-Implant Diseases and Conditions 2017 [19] were used for periodontitis staging. Stage III periodontitis was diagnosed when the following criteria were met: interproximal clinical attachment level (CAL) ≥5 mm in at least two non-adjacent teeth, probing depth (PD) ≥6 mm and radiographic bone loss beyond one-third coronal part of the root. Stage IV periodontitis was diagnosed when in addition to the clinical parameters of stage III, tooth loss ≥5 teeth with an impairment of occlusion and masticatory function were found.’
The compliance was checked every 4 weeks. Line 136-138: ,The compliance with fish oil intake was monitored by calling the patients every four weeks during the medication period to check the bottles back for any possible remaining oil. Then, patients received required bottles of the fish oil for subsequent four weeks.’
There is only one examiner (N.L.) who was calibrated on the different cohort of patients with the use of another experienced periodontist as a rater. An appropriate information has been added to the lines 149-150: ’The study examiner (N.L.) participated in a calibration exercise and the standard error of measurement was calculated.’
Salivary samples were taken at baseline and 3 months. Corrected line 156.
Line 187: ‘comparisons between the test and control groups were performed using the unpaired Students t-test’.
Table 1. p value has been provided.
Results:
Table 3 has been merged with Table 2.
Lines 210-211 corrected: ‘BOP scores showed higher reduction at 3 months in the patients receiving omega-3 PUFAs compared to the control group (p < 0.05).’
Table 4 (now Table 3) has been modified into sections: Cytokines, Chemokines, Growth factors. Different font colors mark pro- and anti-inflammatory factors and different types of chemokines. We believe that all the analyzed factors should be shown in the paper, because this is the first thorough examination of salivary mediator changes before and after SRP. Therefore, we did not remove those that were not statistically significantly different before and after treatment.
Indeed, no statistically significant intragroup changes of concentrations of CCL22 and FGF2 were found in the test group after treatment, thus it is difficult to attribute the difference between the groups at 3 months solely to the effect of omega-3 PUFAs. Appropriate changes were made in Result lines 240-247: ‘Further, we found a significantly higher concentration of fibroblast growth factor (FGF) 2 in saliva in the test group vs. control group at 3 months. This was a result of statistically insignificant raise of FGF2 in the test group (p=0.07) and its unchanged level in the control group (p=0.49) after treatment (Table 3). FGF2 is a multipotent factor responsible for angiogenesis, keratinocyte organization, and wound healing processes. Finally, we detected different tendency in granulocyte-colony stimulating factor (G-CSF) concentrations between the groups. In the test group, G-CSF decreased at 3 months (p=0.02), whereas in the control group it increased, but the change was statistically insignificant (p=0.09) (Table 3).’, and in Discussion lines 363-364: ‘In this study we also found that an intake of fish oil rich in omega-3 PUFAs slightly promoted an increase of salivary concentrations of FGF2.
Discussion:
Tetracyclines and bisphosphonates are not obviously NSAIDs. Corrected line 368: ,Many medications were proposed in this context like non-steroidal anti-inflammatory drugs, tetracyclines or bisphosphonates which may be delivered locally or systemically [9,50,51].’
Minor comments:
Manuscript has been checked for grammar, spelling, style and abbreviations.
Round 2
Reviewer 2 Report
No further comments
Author Response
Thank you once more for your review comments.